# Validation of the HFA-ICOS Score for Carfilzomib-Induced Cardiotoxicity in Multiple Myeloma: A Real-Life Perspective Study

**DOI:** 10.3390/cancers17142353

**Published:** 2025-07-15

**Authors:** Anna Astarita, Giulia Mingrone, Lorenzo Airale, Anna Colomba, Cinzia Catarinella, Marco Cesareo, Fabrizio Vallelonga, Arianna Paladino, Giulia Bruno, Dario Leone, Francesca Gay, Sara Bringhen, Franco Veglio, Alberto Milan

**Affiliations:** 1Division of Internal Medicine, University of Torino, 10126 Turin, Italy; anna.astarita@unito.it (A.A.); lorenzo.airale@unito.it (L.A.); anna.colomba@unito.it (A.C.); cinzia.catarinella@unito.it (C.C.); marcoriccardo.cesareo@unito.it (M.C.); franco.veglio@unito.it (F.V.); alberto.milan@unito.it (A.M.); 2Division of Internal Medicine, Candiolo Cancer Institute, FPO-IRCCS, 10060 Candiolo, Italy; giulia.mingrone@unito.it (G.M.); arianna.paladino@ircc.it (A.P.); giulia.bruno@ircc.it (G.B.); dario.leone@ircc.it (D.L.); 3Division of Hematology, University of Torino, 10126 Turin, Italy; francesca.gay@unito.it (F.G.); sbringhen@cittadellasalute.to.it (S.B.)

**Keywords:** cardiotoxicity, risk prediction model, HFA-ICOS score, carfilzomib, multiple myeloma

## Abstract

This study looks to investigate the performance of the HFA-ICOS risk score in predicting adverse cardiovascular events during Carfilzomib in multiple myeloma patients beyond controlled trials. In a clinical setting, 169 patients with multiple myeloma were divided into classes of risk according to the HFA-ICOS score and followed for a mean of 11.2 months. The incidence of events was high (52.7% of the population) during K therapy. By the results, the HFA-ICOS score did not discriminate between patients at low, medium and high risk for events, showing a limited power to predict the risk of events in our population.

## 1. Introduction

Multiple myeloma (MM) accounts for 1% of neoplastic diseases and is the second most common disease among hematologic malignancies [1], typically affecting elderly populations. In these patients, cardiovascular adverse events (CVAEs) represent the most common complications as a consequence of a high prevalence of coexistent cardiovascular comorbidities, of the increased risk due to MM disease itself, and as a side effect of anti-myeloma treatments [2]. It has been demonstrated that the incidence of CVAEs is higher in patients with MM than in non-MM patients (60.1% vs. 54.5%), and, owing to population aging, the incidence of both MM and cardiovascular diseases is increasing [3]. Carfilzomib (K) therapy, an irreversible second-generation proteasome inhibitor included in different combined treatments, is considered the backbone for patients newly diagnosed and relapsed/refractory with MM, in view of a proven cardiovascular toxicity [4,5,6,7]. A variety of CVAEs were observed during therapy, including heart failure, arrhythmias, acute coronary syndromes, sudden cardiac death and hypertensive events [7,8,9]. Despite the role of prevention and the early identification of patients at higher risk for events being pivotal in patients treated with K, no validated management protocols on cardiovascular risk assessment and follow-up are available. The lack of sufficient data on the main predictors of CVAEs, the heterogeneity of the study protocols and limited data beyond controlled trials in real-life conditions pose a relevant challenge for clinicians. The working group of the Joint Heart Failure Association of the European Society of Cardiology (ESC) and the International Cardio-Oncology Society (HFA-ICOS) have proposed a pretreatment risk assessment tool to stratify cardiovascular risk in cancer patients prior to starting cancer therapies, including K [10,11]. By the application of the score, patients were assigned to separate risk levels prior to starting PI therapies. However, the efficacy of the HFA-ICOS risk assessment tool, specifically for K therapy, has yet to be validated in real-world conditions [12]. The aim of this study was to test the efficacy of the HFA-ICOS tool in a cohort of multiple myeloma patients scheduled for K therapy and prospectively followed, evaluating its role in determining pre-treatment risk for CVAEs.

## 2. Materials and Methods

The study was conducted according to the guidelines of the Declaration of Helsinki and approved by the Ethics Committee of the ‘Città della Salute e della Scienza’ Hospital of Turin (Protocol Number 0038655). This cohort prospective study was conducted at the third-level Hypertension Unit in collaboration with the Myeloma Unit of ‘Città della Salute e della Scienza’ Hospital in Turin, Italy. Patients affected by MM who had an indication for K therapy regimens were consecutively enrolled. Patients previously treated with K, affected by other hematologic diseases, or with cardiac amyloidosis (assessed by end-organ biopsy or cardiac magnetic resonance imaging) were excluded. Furthermore, patients who did not experience cardiovascular events but still had ongoing K treatment at the end of this study were excluded because their outcome was not defined (event versus no event).

### 2.1. Baseline Assessment

In accordance with the recommendations [10,11], the assessment of the baseline cardiovascular risk profile in MM patients before starting K included medical history, office blood pressure (BP) measurements, 12-lead ECG, complete blood tests (including natriuretic peptide and troponins when possible), and trans-thoracic echocardiography (TTE) comprehensive of global longitudinal strain assessment (GLS). In addition, the ambulatory blood pressure monitoring (ABPM) and the estimation of arterial stiffness through pulse wave velocity (PWV) were performed. Details of the study methodology have been reported previously [9]. During the first visit, patients with either office or out-of-office blood pressure values in the high/normal range and those with arterial hypertension, according to the ESH recommendations [13], were advised to start anti-hypertensive treatment or to optimize the previous anti-hypertensive therapy in order to obtain blood pressure control. The type of K-based regimen, timing and dosing were decided by the hematologists.

### 2.2. Application of the HFO-ICOS Score

At baseline, the HFO-ICOS score was calculated considering pre-existing cardiovascular diseases, pre-treatment cardiac biomarkers (if measured), demographic and co-existing medical conditions recognized as cardiovascular risk factors, previous cardiotoxic cancer treatment and lifestyle-related cardiovascular risk factors, as indicated [10,11]. The baseline risk level was derived from the summary of the different variables, as recommended: patients with no risk factors were classified as ‘low risk’, patients with one or more very high risk factors were ‘very high’, and patients with one or more high risk factors were ‘high risk’. Patients with medium risk factors were categorized according to the weight of the medium risk factor as medium 1 or medium 2: patients with one medium 1 risk factor only were ‘low risk’, patients with a single medium 2 risk factor or more than one medium 1 risk factor with points totaling 2–4 were ‘medium risk’, and patients with several medium risk factors with points totaling 5 or more points were ‘high risk’.

### 2.3. Follow-Up

Follow-up assessment consisted of three- to six-month visits (and/or at the time of any suspected CVAE), including the evaluation performed at the baseline visit, except for the ABPM and PWVs, which were performed if needed. Type and incidence of CVAEs were checked during visits, through periodic review of patients’ electronic reports and by phone interviews. CVAEs were recorded during the active K therapy and graded according to the ESC cardio-oncology recommendation. All types of CVAEs were considered (see Appendix B for definitions). Cardiovascular events considered were acute coronary syndromes, heart failure (HF), arrhythmias, typical chest pain, syncope, sudden death, new onset left ventricular dysfunction—defined as reduction in left ventricular ejection fraction—and/or relative decline of GLS value from baseline, according to the current guidelines on management and prevention of cardiotoxicity [11]. Hypertensive events included a new diagnosis of arterial hypertension or the worsening of known arterial hypertension, uncontrolled hypertension prior to K-infusion, uncontrolled hypertension following K-infusion, masked hypertension, hypertensive urgency, and hypertensive emergency, according to the current guidelines [13]. The events were also classified according to the Common Terminology Criteria for Adverse Events (CTCAE), version 5.0, of the National Cancer Institute. Patients were followed until the end of K therapy.

### 2.4. Statistical Analysis

The difference in baseline parameters between patients who experienced CVAEs and patients who did not experience CVAEs was investigated by the chi-square test/Fisher’s exact test for categorical variables and by the unpaired *t*-test/Mann–Whitney test for continuous variables, as appropriate. A two-sided *p*-value less than 0.05 was used as the level of statistical significance. The univariate binary logistic regressions were used to investigate the parameters and levels of risk of the hazard ratio of each variable for CVAEs. The analysis was performed using dedicated software (R: A Language and Environment for Statistical Computing, software version 4.0.0 for Mac OSX, R Core Team; Vienna, Austria).

## 3. Results

### 3.1. Baseline Parameters and Cardiovascular Risk Factors

Between January 2015 and March 2023, 177 of the 210 patients screened for eligibility met the inclusion criteria. Of these, eight patients were still undergoing K therapy and did not experience a CVAE by the end of the study, and they were excluded (Figure 1).

The median age was 70 years (SD 63.0; 74.2), and 77 (45.6%) were male. Among individual risk factors, tobacco use and known arterial hypertension were the most common, with 90 (53%) and 80 (47%) patients, respectively. A large prevalence of pre-existing subclinical organ damage was observed: 30 (18.8%) had left ventricular hypertrophy, 33 (23.1%) had a GLS impairment ≥20%, and 49 (32.2%) had a rise of pulse wave velocity value ≥9 m/s. Baseline characteristics are summarized in Table 1. Furthermore, the population of patients receiving K is confirmed to be a high cardiovascular risk group, characterized by a high prevalence of treated arterial hypertension and comorbidities such as diabetes and dyslipidemia (Table 1). The median duration of K therapy was 16.2 months. The most common regimens were KRD—a combination protocol with lenalidomide and dexamethasone (60 patients; 39.6%)—and KD, a protocol including dexamethasone (54 patients; 35.5%). The K regimens of the enrolled patients are detailed in Appendix A.

### 3.2. Stratification by the HFA-ICOS Score

The assessment of baseline cardiovascular risk before starting K-therapy was based on the application of the HFA-ICOS score [13]. As recommended, the risk level was calculated for each patient from the summary of the baseline risk factors, obtaining four levels of estimated risk for CVAEs: low, medium, high and very high risk (Table 2).

A total of 20 (11.8%) patients were classified as ‘low risk’, 65 (38.5%) as ‘medium risk’, 77 (45.6%) as ‘high risk’ and 7 (4.1%) as ‘very high risk’. In addition, each baseline parameter was compared between the levels of risk (Table 3). Among baseline clinical parameters, the median age, the BMI, known arterial hypertension, previous ischemic heart disease and stroke had a larger prevalence in the levels at higher risk. No differences were found in parameters related to blood pressure profile and in echocardiographic variables, except for the left ventricular mass, which increased at higher levels of risk, as for the pulse wave velocity.

### 3.3. Incidence of CVAEs During K Therapy

All patients received at least one dose of K and were followed for a mean of 11.2 (5.23; 23.8) months. A total of 89 patients (52.7%) experienced at least one CVAEs. A total of 36 patients (21.3%) had at least one cardiovascular adverse event at a median time of 8.33 (SD 3.43; 20.4) months from the initiation of K therapy: heart failure and arrhythmias had the higher incidence (10 and 12 patients, 5.9% and 7.1%, respectively), and one (0.6%) sudden death occurred after 22 months. Three patients had a new LVEF reduction <40%, two had a new LVEF reduction by ≥10 percentage points to an LVEF of 40–49% and a new relative decline in GLS by >15% from baseline, and six (3.4%) had a new decline in GLS > 15% from baseline value with LVEF > 50% (Table 4). A total of 77 patients (45.6%) experienced at least one hypertension-related event. New onset/worsened hypertension and pre-infusion uncontrolled hypertension had the highest incidence (67 and 52, 39.6% and 30.7%, respectively). Five patients (3%) experienced a hypertension emergency. Despite the high number of observed events, the incidence of severe events (CTACE grade ≥ 3) was low. Among the 169 patients, only five required treatment discontinuation due to cardiovascular toxicity.

Comparing the incidence of both cardiovascular and hypertensive events among the different levels of risk, no statistically significant differences were found between groups (Figure 2); the incidence of cardiovascular and hypertensive events appears to be comparable across the different risk levels. Of interest, among the patients classified as high risk according to the HFA-ICOS score, none experienced an event of severity grade ≥3 according to the CTCAE scale.

### 3.4. Accuracy of HFO-ICOS Score and the Role of Other Baseline Parameters as Predictors of CVAEs

The accuracy of the HFO-ICOS in predicting CVAEs during K therapy in our population showed an area under the curve of only 0.557. However, the accuracy of the score was increased by the integration of other baseline parameters. In particular, the presence at baseline of a systolic blood pressure value ≥130 mmHg and of a pulse wave velocity ≥9 m/s was demonstrated to increase the AUC of the score to 0.684 and 0.681 (*p* = 0.03; *p* = 0.038), respectively. Other parameters, such as left ventricular hypertrophy, global longitudinal strain and blood pressure variability, were shown not to increase the accuracy in a statistically significant way (Figure 3A). Conversely, the accuracy of the model further improved by the integration of both parameters, obtaining an AUC of 0.736 (*p* = 0.002), as shown in Figure 3B.

The role of other baseline parameters in predicting future CVAEs was assessed. Parameters such as office systolic and diastolic blood pressure, 24 h systolic blood pressure, day systolic blood pressure and blood pressure variability were significantly associated with future CVAEs. Similarly, parameters of increased arterial stiffness and left ventricular mass were associated with an increased risk of CVAEs. When using the HFO-ICOS score as a continuous variable, the increase in risk levels estimated at baseline was not a predictor for events (Table 5).

## 4. Discussion

Cardiotoxicity induced by protocols containing Carfilzomib poses two major concerns: the morbidity and mortality induced by the cardiovascular event itself and the early discontinuation of the first-line chemotherapy for the hematologic disease, leading to a worse outcome. Despite the inference about cardiotoxic burden related to proteasome inhibitors, the lack of sufficient data in real-world clinical settings and the heterogeneity in detecting and managing cardiovascular events represent significant challenges for clinicians [12]. In a context of profound need for definite recommendations in the field, the working group of the Joint Heart Failure Association of the European Society of Cardiology (ESC) and the International Cardio-Oncology Society (HFA-ICOS) published a risk assessment tool to risk stratify oncology patients prior to receiving therapy with proteasome inhibitors [10]. By the cumulative risk derived by the combination of pretreatment patients and therapy-related factors, these guidelines propose a summary score used to classify patients as low, medium, high and at very high risk for cardiovascular events. However, the effectiveness of the HFA-ICOS score is unclear and requires validation in real-life settings. In our study, we have tested, for the first time, the effectiveness of the HFA-ICOS score in predicting cardiovascular adverse events in a cohort of multiple myeloma patients undergoing protocols including K therapy.

From the results, multiple myeloma patients were confirmed to be a population at medium–high cardiovascular risk, exemplified by a substantial proportion of cardiovascular risk factors and of pre-existing subclinical cardiac (18.8% of left ventricular hypertrophy, 23% of GLS impairment ≥−20 %) and vascular organ damage (32% had a rise of pulse wave velocity value ≥−9 m/s). During a median of 11.2 months of follow-up from K starting, the incidence of CVAEs was high: 89 (52.7%) patients experienced at least one event. Adverse cardiovascular events occurred in 21.3% of patients with various degrees of cardiac toxicity; one fatal cardiotoxicity. One of the important features of this study, as in our previous report [7], was the exploration of the rates of mild cancer therapy-related cardiac dysfunction (3.6% of patients), defined as a decrease in global longitudinal strain even when unaccompanied by an LVEF < 50%, according to the latest consensus statement from the International Cardio-Oncology Society [11]. Nearly half of the patients experienced at least one hypertensive event, and 3% had a hypertensive emergency. This incidence was higher than the results observed in preclinical trials [14], in which the rate of CVAEs ranged from 18% to 22% [15,16], but consistent with the findings of studies conducted in curative settings [7], as in Cornell’s study [8], in which the incidence of cardiovascular adverse events during K therapy was observed in about 50% of the population. With regard to hypertensive events, Carfilzomib confirms its role in contributing to blood pressure elevation. This effect may, however, be partially influenced by the concomitant use of dexamethasone in the various treatment regimens. Nonetheless, in the previous study by Astarita et al. [7], which compared the incidence of cardiovascular events between two treatment regimens based on different doses of Carfilzomib—KD (Carfilzomib–dexamethasone, target dose of K: 56 mg/m^2^) versus KRD (Carfilzomib–lenalidomide–dexamethasone, target dose of K: 27 mg/m^2^)—both all-type cardiovascular events and hypertensive events occurred more frequently in patients receiving KD compared to those receiving KRD. These findings suggest a primary role of Carfilzomib in the development of such events, even at an equivalent dose of dexamethasone. Nevertheless, although a substantial number of cardiovascular and hypertensive events were observed, the incidence of severe events (CTACE grade ≥3) was low. Among patients who experienced grade ≥3 events, only five required discontinuation of Carfilzomib therapy. In fact, the majority of CVAEs were transient and of low severity. In the remaining cases, treatment with Carfilzomib was continued after either intensification of anti-hypertensive therapy or optimization of cardioprotective management.

By applying the HFA-ICOS risk score at baseline, 11.8% of patients were classified as ‘low risk’, 38.5% as ‘medium risk’, 45.6% as ‘high risk’ and 4.1% as ‘very high risk’ for CVAEs during K therapy. Consequently, about half of the patients were classified as having high or very high risk. Analyzing the characteristics of each group, parameters such as median age, BMI, arterial hypertension, previous ischemic heart disease, stroke, left ventricular mass, and pulse wave velocity were more prevalent in groups at higher risk. However, these parameters, with the exception of the pulse wave velocity, were themselves risk factors considered by the model. In contrast, the groups did not differ for parameters related to office and out-of-office blood pressure profiles or for the other echocardiographic variables. Furthermore, the HFA-ICOS score did not show a good correlation with the incidence of CVAEs during K therapy in our population, even when considered as a continuous variable. During the follow-up, no differences were observed for the incidence of cardiovascular events, hypertensive-related events, or any CVAEs between the low, medium, high and very high risks at baseline (*p* > 0.05 for all comparisons). As shown in Figure 2, in a population of 169 patients, the score identified only 7 patients as being at very high risk. While it is true that four of these patients (57%) experienced at least one cardiovascular event, the lower-risk groups—which also included a larger number of patients—showed a comparable incidence of events. Therefore, the HFA-ICOS score showed a low sensitivity and did not discriminate between patients at low and high risk for events in a clinical setting. Of interest, among the patients classified as high risk according to the HFA-ICOS score, none experienced an event of severity grade ≥3 according to the CTCAE scale. This finding suggests a limited clinical applicability of the score within our study population. These limits of applicability of the score in our clinical setting could be related to the studies that support the score itself. Indeed, with the exception of the study of Cornell et al. [8], the HFA-ICOS is based on expert opinions, studies based on small sample sizes [17], not including K treatment [18,19] and preclinical studies [16] with significant heterogeneity in the definition of cardiovascular events and limited validation to date. It follows a limited applicability in the real-life world, as highlighted by other clinical studies. In women with HER2+ treated with trastuzumab, two retrospective studies, Suntheralingam et al. [20] and Battisti et al. [21], demonstrated that the HFA-ICOS risk score did not adequately identify patients at low risk for cardiac dysfunction. For overall cardiotoxicity related to cancer treatment, in Cronin’s study [22], the HFA-ICOS showed a low sensitivity with a moderate power in predicting CVAEs in a cohort of women with HER2+ breast cancer. Similarly, Tini et al. [23], in two cohorts of breast cancer women treated with anthracyclines and anti-HER2, concluded that patients classified at medium–high risk using the HFA-ICOS score were not associated with the occurrence of cardiac dysfunction.

Of interest, we investigated the performance of the HFA-ICOS score when integrated with other baseline parameters. In particular, the presence at baseline of a systolic blood pressure value ≥130 mmHg and of a pulse wave velocity ≥9 m/s demonstrated to increase the accuracy of the score, obtaining the maximum accuracy including both the parameters in the model (AUC of 0.736). These findings are consistent with other reports [7] that identified the systolic blood pressure—and, in general, parameters related to in-office and out-of-office blood pressure—and the pulse wave velocity as predictors of CVAEs during K therapy. Consequently, it is conceivable that the inclusion of these variables in the HFA-ICOS score could improve the performance of the score in detecting high-risk patients in real-life settings.

The current study has some limitations. Firstly, the experience of a single center limits the generalizability of the results, which should be confirmed by larger studies with a multicenter design. Moreover, our study lacks some data points and routine follow-up information that are included in the HFA-ICOS score, which introduces potential biases affecting its applicability. In particular, the incidence of venous thrombosis and pulmonary embolism was not assessed, and cardiac biomarkers (troponin and BNP or NT-proBNP) were not routinely measured for all patients. Similarly, we did not include patients affected by cardiac amyloidosis (exclusion criteria), one of the risk factors considered by the HFA-ICOS score. These limitations may lead to incorrect patient stratification as indicated by the score.

## 5. Conclusions

Our study evaluated for the first time the performance of the HFA-ICOS risk score in predicting cardiovascular adverse events, classified according to the latest consensus statement from the International Cardio-Oncology Society, in a cohort of multiple myeloma patients treated with K therapy in a real-world clinical setting and prospectively followed. The application of the HFA-ICOS risk score at baseline in our population showed limited ability to discriminate patients at low, medium, and high risk for cardiovascular and hypertensive events during K therapy. Indeed, risk groups exhibited a similar incidence and severity of both cardiovascular and hypertensive events. Of interest, the integration of additional parameters into the HFA-ICOS score, such as systolic blood pressure and pulse wave velocity, has been shown to improve the score’s performance. This study represents a first step in understanding the applicability of the score outside controlled clinical trials, and future research on larger, multicenter cohorts is needed to validate these findings.

## Figures and Tables

**Figure 1 cancers-17-02353-f001:**
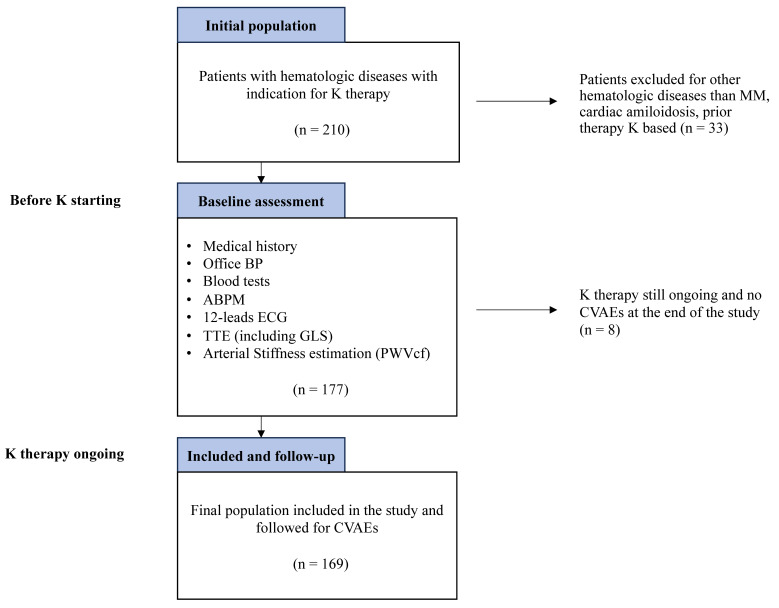
Flowchart of the study protocol.

**Figure 2 cancers-17-02353-f002:**
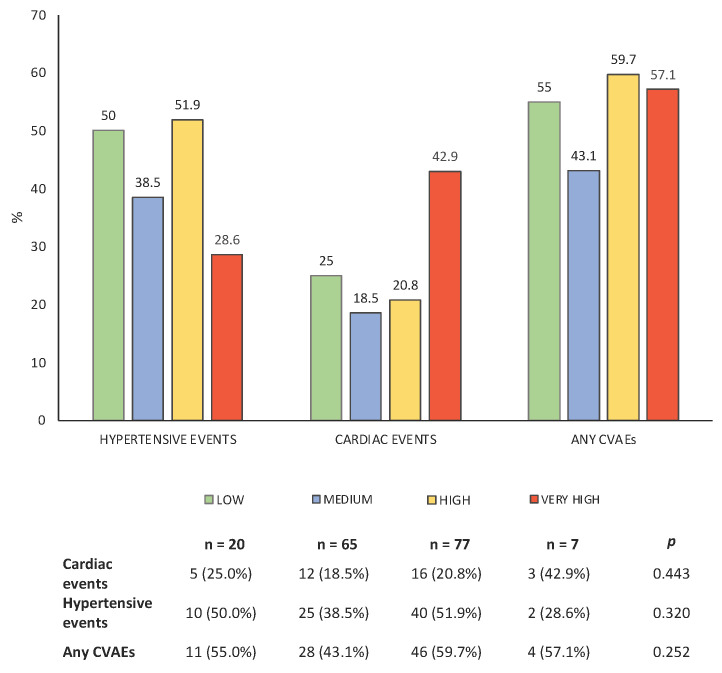
Incidence of CVAEs by baseline risk groups of HFO-ICOS score.

**Figure 3 cancers-17-02353-f003:**
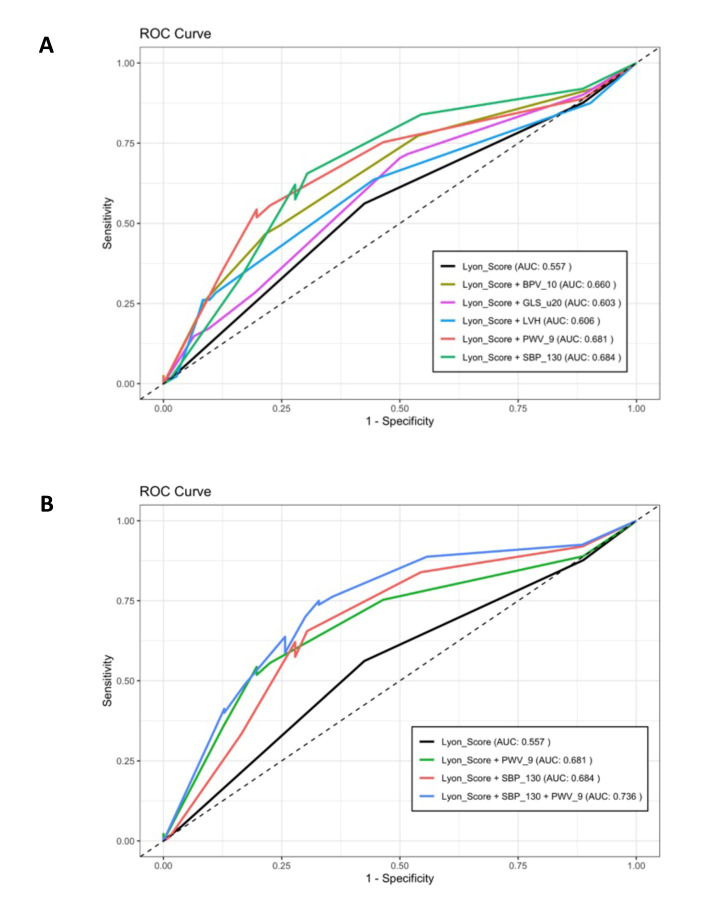
Accuracy of the HFO-ICOS in predicting CVAEs. (**A**) ROC curve of HFO-ICOS score (black line) and of HFO-ICOS comprehensive of blood pressure variability > 10 (dark green line), global longitudinal strain (GLS) > −20% (pink line), left ventricular hypertrophy (blue line), pulse wave velocity ≥ 9 m/s, and systolic blood pressure (SBP) ≥ 130 mmHg (green line). (**B**) ROC curve of HFO-ICOS score (black line), HFO-ICOS score comprehensive of pulse wave velocity ≥ 9 m/s (green line), HFO-ICOS score comprehensive of systolic blood pressure (SBP) ≥ 130 mmHg (red line) and HFO-ICOS score comprehensive of both systolic blood pressure (SBP) and pulse wave velocity ≥ 9 m/s (blue line).

**Table 1 cancers-17-02353-t001:** Baseline parameters of the population.

Parameter	N = 169 (100%)
*General*	
Age, y	70.0 (63.0; 74.2)
Male sex	77 (45.6%)
BMI, kg/m^2^	26.7 (23.9; 30.3)
*Individual CV risk factors and comorbidities*	
Tobacco use (prior/current)	90 (53.3)
Obesity (BMI ≥ 30)	49 (29)
Known arterial hypertension	80 (47.3)
Diabetes mellitus	16 (9)
Chronic renal failure (eGFR < 60 mL/m)	41 (26.3)
Ischemic heart disease	4 (2.3)
Atrial fibrillation	5 (2.9)
Dyslipidemia	25 (14.8)
Previous stroke	3 (1.8)
*Office BP values*	
SBP, mmHg	128 (116; 141)
DBP, mmHg	76.0 (70.0; 84.0)
*ABPM* ^1^ *, mean (SD)*	
Daytime SBP, mmHg	124 (116; 132)
Daytime DBP, mmHg	74.0 (69.0; 80.0)
Daytime MBP, mmHg	92.0 (86.0; 96.5)
24 h SBP, mmHg	120 (112; 129)
24 h DBP, mmHg	71.0 (65.0; 76.5)
24 h MBP, mmHg	88.0 (82.0; 93.0)
Nighttime SBP, mmHg	109 (102; 120)
Nighttime DBP, mmHg	62.0 (57.0; 69.0)
Night MBP, mmHg	78.0 (73.0; 86.0)
Blood pressure variability	9.00 (7.0; 10.0)
*Echocardiographic parameters* ^2^	
LAVi, ml/m^2^	49.0 (38.4; 61.2)
LVMi, g/m^2^	86.4 (74.4; 102.0)
LVH	30 (18.8)
Diastolic dysfunction	9 (6.4)
LVEF, %	62.2 (58.3; 65.2)
Stroke volume, mL/m^2^	47.4 (39.1; 57.9)
GLS value, %	−21.50 (−23.40; −20.10)
GLS value ≥ −20 %	33 (23.1)
*Arterial stiffness evaluation* ^3^	
cfPWV value, m/s	8.00 (7.0; 9.0)
cfPWV value ≥ 9 m/s	49 (32.2)
*Oncological history*	
Median MM disease duration, months	11.2 (5.23; 23.8)
*Line of K therapy*	2.85 (1; 9)

Mean values estimated in ^1^ 152 patients; ^2^ 165 patients; ^3^ 153 patients. BMI = body mass index; eGFR = estimated glomerular filtration rate; ABPM = ambulatory blood pressure monitoring; SBP = systolic blood pressure; DBP = diastolic blood pressure; MBP = mean blood pressure; LVH = left ventricular hypertrophy; BP = blood pressure; LAVi = left atrium volume index; LVMi = left ventricular mass index; LVH = left ventricular hypertrophy; GLS = global longitudinal strain; cfPWV = carotid–femoral pulse wave velocity; MM = multiple myeloma; SD = standard deviation.

**Table 2 cancers-17-02353-t002:** Stratification by the HFA-ICOS score of MM patients scheduled for K at baseline.

	Patients (n)	Score
*Previous* *cardiovascular diseases*		
Heart failure or cardiomyopathy	2 (1.2)	Very High
Prior proteasome inhibitor cardiotoxicity	0 (0)	Very High
Venous thrombosis (DVT or PE)	0 (0)	Very High
Cardiac amyloidosis	na	Very High
Arterial vascular disease (IHD, PCI, CABG, stable angina, TIA, stroke, PVD)	7 (4.1)	Very High
Prior immunomodulatory CV toxicity	0 (0)	High
Baseline LVEF < 50%	5 (3)	High
Borderline LVEF 50–54%	5 (3)	Medium 1
Arrhythmia	5 (3)	Medium 2
Left ventricular hypertrophy ^1^	30 (17.8)	Medium 1
*Cardiac biomarkers* ^2^		
Elevated baseline troponin	0 (0)	Medium 2
Elevated baseline BNP or NT-proBNP	2 (1.2)	High
*Demographic and cardiovascular risk factors*		
Age ≥ 75 years	40 (23.7)	High
Age 65–74 years	79 (46.7)	Medium 1
Arterial hypertension	80 (47.3)	Medium 1
Diabetes mellitus	16 (9.5)	Medium 1
Hyperlipidemia	25 (14.8)	Medium 1
Chronic kidney disease	40 (23.7)	Medium 1
Family history of thrombophilia	na	Medium 1
*Previous cardiotoxic cancer treatment*		
Prior anthracycline exposure	34 (20.1)	High
Prior thoracic spine radiotherapy	7 (4.1)	Medium 1
*Current myeloma treatment*		
High-dose of dexamethasone > 160 mg/month	0 (0)	Medium 1
*Lifestyle risk factors*		
Current smoker or significant smoking history	90 (53.3)	Medium 1
Obesity (BMI > 30 kg/m^2^)	49 (29)	Medium 1

Mean values estimated in ^1^ 176 patients; ^2^ 14 patients. BMI = body mass index; BNP = brain natriuretic peptide; CABG = coronary artery bypass graft; DVT = deep vein thrombosis; IHD = ischemic heart disease; LVEF = left ventricular ejection fraction; NT-proBNP = N-terminal pro-brain natriuretic peptide; PCI = percutaneous coronary intervention; PE = pulmonary embolism; PVD = peripheral vascular disease; TIA = transient ischemic attack; na = not applicable.

**Table 3 cancers-17-02353-t003:** Comparison of baseline parameters between the levels of risk based on HFO-ICOS score stratification.

Parameter	Levels of Risk					
	1. Low N = 20	2. Medium N = 65	3. High N = 77	4. Very High N = 7	*p*	Comparison
*General*						
Age, y	60.0 (54.0; 65.0)	69.0 (63.0; 72.0)	75.0 (68.0; 77.0)	68.0 (66.5; 68.5)	<0.001	1 vs. 2/3/4; 2 vs. 3
Male sex	8 (40.0)	27 (41.5)	38 (49.4)	4 (57.1)	0.677	
BMI, kg/m^2^	24.8 (21.8; 27.0)	28.1 (25.2; 31.6)	26.4 (23.7; 30.1)	27.6 (27.3; 31.4)	0.012	1 vs. 2
*Individual CV* *risk factors*						
Tobacco use (prior/current)	7 (35.0)	37 (56.9)	41 (53.2)	5 (71.4)	0.275	
Obesity (BMI ≥ 30)	0 (0.0)	22 (33.8)	24 (31.2)	3 (42.9)	0.005	1 vs. 2/3/4
Known arterial hypertension	2 (10.0)	30 (46.2)	42 (54.5)	6 (85.7)	<0.001	1 vs. 2/3/4
Diabetes mellitus	0 (0)	5 (7.7)	10 (13.0)	1 (14.3)	0.253	
Chronic renal failure (eGFR < 60 mL/m)	92.9 (77.3; 106)	80.7 (64.7; 99.0)	67.8 (54.8; 93.5)	81.5 (68.3; 89.0)	0.033	1 vs. 3
Ischemic heart disease	0 (0)	0 (0)	0 (0)	4 (57.1)	<0.001	4 vs. 1/2/3
Atrial fibrillation	0 (0)	1 (1.5)	3 (3.9)	1 (14.3)	0.277	
Dyslipidaemia	0 (0)	13 (20)	12 (15.6)	0 (0)	0.099	
Previous stroke	0 (0)	0 (0)	0 (0)	3 (42.9)	<0.001	4 vs. 1/2/3
*Office BP values,* *mmHg*						
SBP	120 (114; 128)	128 (119; 141)	130 (119; 142)	122 (104; 131)	0.081	
DBP	74.5 (70.0; 80.0)	77.5 (70.8; 87.2)	75.0 (69.0; 82.0)	79.0 (65.5; 81.5)	0.501	
*ABPM,* *mmHg*						
Daytime SBP	120 (114; 130)	126 (118; 133)	124 (116; 132)	114 (107; 116)	0.095	
Daytime DBP	74.0 (70.0; 80.5)	76.5 (69.0; 82.0)	73.0 (68.0; 78.0)	66.5 (60.5; 72.5)	0.066	
Daytime MBP	88.5 (84.8; 96.8)	93.0 (88.8; 99.0)	91.0 (86.0; 95.5)	84.5 (77.0; 92.0)	0.104	
24 h SBP	114 (107; 124)	122 (114; 129)	120 (112; 129)	108 (104; 115)	0.101	
24 h DBP	70.0 (65.2; 76.8)	73.0 (67.0; 77.5)	70.5 (64.8; 75.0)	64.5 (58.0; 71.0)	0.250	
24 h MBP	85.0 (79.5; 92.2)	89.0 (84.0; 93.0)	89.0 (82.0; 93.0)	81.0 (72.8; 90.0)	0.334	
Nighttime SBP	104 (96.8; 112)	110 (102; 121)	112 (103; 121)	99.5 (95.5; 109)	0.093	
Nighttime DBP	59.5 (56.2; 66.0)	63.5 (59.0; 69.0)	63.0 (58.0; 70.0)	60.0 (53.5; 70.2)	0.484	
Night MBP,	73.5 (71.2; 80.8)	79.0 (74.0; 86.0)	79.0 (74.0; 88.8)	74.5 (69.2; 82.8)	0.147	
Blood pressure variability	8.00 (7.00; 9.00)	9.00 (7.50; 11.0)	8.50 (7.00; 10.0)	8.50 (6.50; 11.2)	0.597	
*Echocardiographic* *parameters*						
LAVi, mL/m^2^	39.0 (33.3; 65.0)	49.4 (41.2; 63.0)	48.8 (38.3; 58.1)	57.7 (49.2; 65.5)	0.341	
LVMi, g/m^2^	74.5 (66.7; 84.8)	82.7 (70.2; 95.7)	90.4 (76.9; 104)	115 (86.1; 119)	0.002	1 vs. 3/4
LVH,	1 (5.26%)	7 (11.5%)	18 (24.7%)	4 (57.1%)	0.007	1/2 vs. 4
Diastolic dysfunction	0 (0.00%)	4 (7.69%)	4 (5.97%)	1 (16.7%)	0.451	
LVEF, %	60.2 (57.6; 64.3)	62.7 (58.9; 65.6)	62.2 (58.6; 64.7)	60.8 (55.0; 62.2)	0.491	
Stroke volume, mL/m^2^	53.0 (39.7; 60.8)	50.2 (40.7; 61.5)	43.5 (36.1; 53.1)	54.9 (53.5; 68.8)	0.023	
GLS value, %	−21.90 (−23.40; −20.30)	−21.60 (−22.70; −19.60)	−21.60 (−23.90; −20.25)	−21.00 (−21.50; −21.00)	0.707	
GLS value ≥ −20%	2 (11.8%)	15 (28.3%)	15 (22.1%)	1 (20.0%)	0.580	
*Arterial stiffness* *evaluation*						
cfPWV value, m/s	7.00 (6.00; 7.50)	8.00 (7.00; 9.00)	9.00 (7.00; 10.0)	8.00 (7.00; 9.00)	0.001	1/2 vs. 3
cfPWV value ≥ 9 m/s	2 (10.5%)	18 (30.5%)	36 (52.2%)	2 (40.0%)	0.002	1 vs. 3

BMI = body mass index; eGFR = estimated glomerular filtration rate; SBP = systolic blood pressure; DBP = diastolic blood pressure; MBP = mean blood pressure; LAVi = left atrium volume index; LVMi = left ventricular mass index; LVH = left ventricular hypertrophy; LVH = left ventricular hypertrophy; GLS = global longitudinal strain; cfPWV = carotid–femoral pulse wave velocity.

**Table 4 cancers-17-02353-t004:** Incidence of CVAEs during K therapy.

Events	N = 169 (100%)	CTCAE	
		Grade 1–2	Grade ≥ 3
**Cardiovascular adverse events** ^1^	**36 (21.3)**		
ACS (STEMI)	2 (1.2)	0	2
ACS (NSTEMI)	4 (2.4)	0	4
Angina	9 (5.3)	9	0
Heart failure	10 (5.9)	8	2
Arrhythmias	12 (7.1)	11	1
Sudden death	1 (0.6)	na	0
LVEF impairment	5 (3)	0	3
GLS impairment	6 (3.6)	na	na
**Hypertensive events ^1^**	**77 (45.6)**		
New onset/worsened hypertension	67 (39.6)	67	0
Masked hypertension	5 (3)	5	0
K pre-infusion hypertension permissive K infusion	34 (20.1)	27	7
K pre-infusion hypertension not permissive K infusion	18 (10.7)	11	7
K post-infusion hypertension	20 (11.8)	13	7
Hypertensive urgency	5 (3)	0	5
Hypertensive emergency	0 (0)	0	0
**Any CVAEs**	**89 (52.7)**		

^1^ Patients experienced more than one CVAE; hence, the total % amounts to >100. CVAE = cardiovascular adverse event; ACS = acute coronary syndrome; STEMI = ST-elevation myocardial infarction; NSTEMI = non-ST elevation myocardial infarction. CVAEs = cardiovascular adverse events. CTCAE = Common Terminology Criteria for Adverse Events na = not applicable.

**Table 5 cancers-17-02353-t005:** The role of other baseline parameters in predicting CVAEs during K therapy.

Baseline Parameter	Beta	OR (SD)	Wald Test	Valore di p
*General*				
Age	0.02	1.02 (0.98–1.06)	1.12	0.264
Male sex	0.44	1.55 (0.84–2.86)	1.40	0.160
BMI	0.03	1.03 (0.96–1.10)	0.81	0.419
Office BP				
SBP	0.04	1.04 (1.02–1.07)	3.94	<0.001
DBP	0.03	1.03 (1.00–1.06)	2.03	0.042
*ABPM*				
24 h SBP	0.03	1.03 (1.00–1.06)	1.97	0.048
24 h DBP	0.02	1.02 (0.98–1.06)	0.92	0.358
24 h MBP	0.02	1.02 (0.99–1.06)	1.15	0.249
Daytime SBP	0.03	1.03 (1.01–1.06)	2.41	0.016
Daytime DBP	0.02	1.02 (0.99–1.06)	1.21	0.225
Daytime MBP	0.03	1.03 (1.00–1.07)	1.85	0.065
Nighttime SBP	0.01	1.01 (0.98–1.03)	0.42	0.674
Nighttime DBP	−0.00	1.00 (0.96–1.04)	−0.16	0.872
Night MBP	0.00	1.00 (0.97–1.04)	0.08	0.936
Blood pressure Variability	0.18	1.19 (1.07–1.35)	3.06	0.002
*Echocardiographic parameters*				
LAVi	0.00	1.00 (0.98–1.02)	0.17	0.865
LVEF	0.02	1.02 (0.97–1.07)	0.77	0.441
LVMi	0.02	1.02 (1.01–1.04)	2.69	0.007
LVH	1.19	3.29 (1.38–8.77)	2.55	0.011
Diastolic dysfunction	−0.94	0.39 (0.08–1.55)	−1.29	0.199
GLS value	0.10	1.10 (0.97–1.26)	1.47	0.142
GLS value ≥ −20	0.55	1.73 (0.78–4.02)	1.32	0.188
Arterial stiffness evaluation				
cfPWV	0.40	1.49 (1.23–1.84)	3.87	<0.001
cfPWV ≥ 9 m/s	1.58	4.84 (2.38–10.32)	4.23	<0.001
*HFA-ICOS score*				
Level of risk	0.24	1.27 (0.85–1.92)	1.15	0.251

SBP = systolic blood pressure; DBP = diastolic blood pressure; MBP = mean blood pressure; LAVi = left atrium volume index; LVEF = left ventricular ejection fraction; LVMi = left ventricular mass index; LVH = left ventricular hypertrophy; GLS = global longitudinal strain; cfPWV = carotid–femoral pulse wave velocity.

## Data Availability

Data are contained within the article.

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
