# Peer review of "Validation of the HFA-ICOS Score for Carfilzomib-Induced Cardiotoxicity in Multiple Myeloma: A Real-Life Perspective Study"

_cancers, 2025, doi:10.3390/cancers17142353_

Round 1
Reviewer 1 Report
Comments and Suggestions for Authors
Astarita, A., et al. reported the clinical significance of the HFA-ICOS score in multiple myeloma patients who received carfilzomib-based therapy. The authors retrospectively analyzed the incidence of carfilzomib-induced cardiovascular toxicity according to the risk classification of the HFA-ICOS score in 169 myeloma patients. They found that 36 patients (21.3%) had at least one cardiovascular event and 77 patients (45.6%) had almost one hypertension related event. Unexpectedly, the incidence of both cardiovascular and hypertensive events was not significantly related with the risk level according to the risk classification of the HFA-ICOS score. Importantly, the presence of the baseline of systolic blood pressure and pulse wave velocity showed to increase the accuracy of the score. Based on these results, they concluded that the HFA-ICOS score did not discriminate between patient risk during the treatment with carfilzomib in multiple myeloma patients. However, there are several concerns regarding the paper that the authors need to clarify.
- What exactly is a regimen that includes carfilzomib? Dosage and schedule of carfilzomib needs to be mentioned. If it includes dexamethasone, it may also affect adverse events in hypertension, which needs to be discussed.
- In Figure 2, the very high-risk group appears to have a predominantly high rate of cardiovascular events, and additional explanations are recommended.
- The results are summarized only in terms of the incidence of cardiovascular events and hypertensive adverse events, which makes it difficult to understand the actual clinical significance of the study. Adverse events should be described by the grade of CTCAE, and the grade 3 or higher adverse events or the rate of discontinuation of carfilzomib treatment should be analyzed more in detail from the clinical point of view.
- The incidence of cardiovascular and hypertensive events in this study is considered to be extremely high, and further discussion of this point is warranted.
- The authors should include a discussion of why the HFA-ICOS score is not useful for prediction of carfilzomib-induced cardiotoxicity in relation of the mechanism of this adverse event.
Reviewer 2 Report
Comments and Suggestions for Authors
The authors studied and validated the HFO-ICOS scoring system on myeloma patients with carfilzomib therapy. They found HFO-ICOS scoring system could not clarify the risks of CV events in these patients.
- the single center study with limited number of patients, long-term of study period, and missing items/not routinely follow-up data of HFA scoring system made bias on the study.
- The patients characteristics, including co-morbidities, which lines of therapy should be discussed.
- The schedule and dosing of carfilzomib and the duration of treatment should be discussed.
- there is no enough information to the readers from the conclusion.
Round 2
Reviewer 1 Report
Comments and Suggestions for Authors
The authors revised the manuscript appropriately according to the reviewers' comments.